# Strategies to Enhance the Membrane-Based Processing Performance for Fruit Juice Production: A Review

**DOI:** 10.3390/membranes13070679

**Published:** 2023-07-20

**Authors:** Kamil Kayode Katibi, Mohd Zuhair Mohd Nor, Khairul Faezah Md. Yunos, Juhana Jaafar, Pau Loke Show

**Affiliations:** 1Department of Process and Food Engineering, Faculty of Engineering, Universiti Putra Malaysia, UPM, Serdang 43400, Selangor, Malaysia; kfaezah@upm.edu.ng; 2Department of Agricultural and Biological Engineering, Faculty of Engineering and Technology, Kwara State University, Malete 23431, Nigeria; 3Laboratory of Halal Science Research, Halal Products Research Institute, Universiti Putra Malaysia, Putra Infoport, UPM, Serdang 43400, Selangor, Malaysia; 4N29a, Advanced Membrane Technology Research Centre (AMTEC), Faculty of Chemical and Energy Engineering, Universiti Teknologi Malaysia, UTM Skudai, Johor Bahru 81310, Johor, Malaysia; juhana@petroleum.utm.my; 5Department of Chemical Engineering, Khalifa University, Abu Dhabi P.O. Box 127788, United Arab Emirates; pauloke.show@ku.ac.ae

**Keywords:** membrane, juice clarification, ultrafiltration, fouling

## Abstract

Fruit juice is an essential food product that has received significant acceptance among consumers. Harmonized concentration, preservation of nutritional constituents, and heat-responsive sensorial of fruit juices are demanding topics in food processing. Membrane separation is a promising technology to concentrate juice at minimal pressure and temperatures with excellent potential application in food industries from an economical, stable, and standard operation view. Microfiltration (MF) and ultrafiltration (UF) have also interested fruit industries owing to the increasing demand for reduced pressure-driven membranes. UF and MF membranes are widely applied in concentrating, clarifying, and purifying various edible products. However, the rising challenge in membrane technology is the fouling propensity which undermines the membrane’s performance and lifespan. This review succinctly provides a clear and innovative view of the various controlling factors that could undermine the membrane performance during fruit juice clarification and concentration regarding its selectivity and permeance. In this article, various strategies for mitigating fouling anomalies during fruit juice processing using membranes, along with research opportunities, have been discussed. This concise review is anticipated to inspire a new research platform for developing an integrated approach for the next-generation membrane processes for efficient fruit juice clarification.

## 1. Introduction

The juice obtained from the fruit, as well as vegetables, comprises vitamins, minerals, carbohydrates, as well as phytochemical compounds. The sensory quality of juice and its nutritional value are usually preserved using various processing techniques while producing juice and maintaining its safety and stability. Juice clarification is an essential phase in producing clear juice to achieve clear beverages and avert sediment accretion during storage [1].

This review specifically focuses on the application of membrane technology for fruit juice clarification. The initial section succinctly discusses fruit juice clarification using conventional thermal evaporation techniques and their associated drawbacks. The following sections provide a brief outlook on membrane processes that clarify various fruit juices. Particular attention is devoted to critical factors that control these techniques’ performances during fruit juices clarification. A final section adequately discusses recent trends in the strategies employed in advancing the performance of membrane processes and fouling mitigation during the clarification of fruit juices (pre-treatment, membrane modification, membrane cleaning, etc.). A dedicated review of essential factors controlling the membrane filtration performance, as well as strategies for mitigating fouling and enhancing membrane separation during fruit juice processing, has not been reported.

The industrial clarification and concentration of fruit juices are typically accomplished via multistep thermal evaporation procedures, in which the water is separated at elevated temperatures. Thermal evaporation is the most convenient technology for fruit juice concentrates in the food industry. However, the major drawbacks of these procedures include elevated-energy demand, color degradation, off-flavor build-up, change in taste, deficiency of aroma compounds in the juice concentrate, and most importantly, decline in nutritional properties owing to thermal impacts. Generally, the conventional clarification techniques utilized in the fruit juice manufacturing industry are comprised of numerous unit processes comprising depectinization (enzymatic treatment), cooling, flocculation (using silica sol, gelatine, bentonite, and diatomaceous earth), decantation, centrifugation, then filtration. These techniques are laborious and time-sapping because the flocculation step demands 6–18 h for sufficient sedimentation [2]. Also, there are some additional shortcomings to using clarifying agents like bentonite. In addition, traditional juice processing requires thermic treatments to achieve shelf-life extension and juice concentration. Unfortunately, thermally treated juice is highly vulnerable to quality deterioration, such as non-enzymatic browning and aroma loss, as well as declines in ascorbic acid and total polyphenols [3].

Given this, the continuously rising demand for superior-quality and healthy beverages by consumers has triggered the interest of scientists to explore various efficient and superior techniques in the production of new juice products via minimum unit processing and heat. Hence, several non-heat-processing techniques, comprising membrane filtration [4], pulsed electric field [5], ozone [6], ultrasound [7], high pressure [8], cold plasma [9], and UV-C light [10], and freeze concentration [11] have been exploited for juice concentration, clarification, as well as preservation that can efficiently maintain the sensory along with nutritional properties of juice [12,13,14]. Membrane separation technology and cryo-concentration (freeze concentration) are the major recent techniques that could be employed to adequately sustain the uniqueness of the fresh fruit in terms of its nutritional value, pristine color, aroma, structural properties as well as stable products. However, the latter technique is undesirable owing to higher energy demand and reduced attainable amount of concentration (40 gTSS/100 g) in comparison with the conventional thermal technique [15].

Membrane technology is more attractive and widely applied to separate solids from liquid in the food and fruit juice industry than conventional techniques, for instance, to clarify fruit juices [16,17]. Modern membrane techniques are extensively applied to separate solids from liquids in the fruit juice production industry [18] and employed in the food industry in the processing of different juices and beverages to produce fresh, natural products of superior quality without heat destruction or chemical by-products [19]. Membrane technology is considered an effective technique for juice clarification and concentration, offering numerous advantages over conventional separation technologies due to its unique properties, including excellent selectivity based on better separation properties, zero thermal stress of treated fluids attributable to moderate working temperatures, zero use of chemical supplements, modular and compact configuration, and minimal energy demand [20]. It also provides an efficient pathway to lower turbidity and exterminates large particles in the fruit juices, with excellent separation performance without undermining juice quality at lower initial investment costs and comparatively minimal energy demand, using facile preparation techniques [21,22].

Membrane processes such as RO, MF, UF, FO, and NF have been extensively applied in the food and fruit industry. UF holds the largest market share among these processes due to superior selectivity and minimal energy [23,24]. MF is widely used to concentrate and separate large particles, including fruit juice clarification and phenolic compounds from plant materials [25]. MF has been applied productively to clarify pomegranate juice and other fruit juices and maintains the chemical characteristics of the juice in comparison with enzymatic processes. Similarly, UF membranes fabricated from polysulfone (PSF) have the largest market share among the membrane processes. They are extensively applied to concentrate the juice and clarify fruit juice by eliminating tannins, microscopic organisms, molds, proteins, yeast, polysaccharides, and colloids, in the fruit juice industry, owing to their superior selectivity, inexpensive, superior film-forming potential, outstanding mechanical properties, and higher chemical and thermal resistance in addition to minimal energy demand [26]. Several reports have indicated that UF and MF membrane processes are the most widely applied for the clarification of various fruit juices among the various membrane processes, while PSF and PES are the most widely used polymeric materials in the fabrication and application of polymeric membrane processes as highlighted in Table 1 and Table 2. However, some studies have also explored the use of ceramic membranes for the clarification of fruit juices (Table 2).

However, the hydrophobic property of polymeric membranes remains the main stumbling block triggering the fouling stemming from the agglomeration of large particles (feed) in the pore channel as well as on the surface of the membrane [27]. The accretion of large particles on the membrane surface declines the juice permeate flux resulting in membrane fouling, which is the prime obstacle to membrane separation [28]. Fouling reduces the flux and separation performance of the product during filtration and increases energy consumption [29]. This behavior justifies the need to elucidate the various controlling factors that could affect the performance of membrane separation efficiency.

**Table 1 membranes-13-00679-t001:** Summary of findings on concentration strength and permeate flux and significant findings during various fruit juices clarifications.

Juice Source	Membrane Type and Operating Conditions	Concentration (°Brix)		Flux (L/m^2^h)		Major Findings	References
		Starting	Final	Starting	Final		
Apple	UF; A:150 cm^2^; TMP: 5.4 bar; Time: 120 & 150	13.0	16.5	26.3	44.6	Total phenolic content increases from 107 to 312.3 (mg GAE/L).	[30]
Keylime, watermelon, kiwifruit	CM; Time: 100 min; TMP: 300, 500, and 700 kPa	11.6	11.8	273	388	Excellent mechanical strength and highly resistant to operating pressures up to 700 kPa with small intrinsic resistance values at 0.023, 0.475, and 0.488.	[31]
Pomegranate	MF; A: 14.6 cm^2^; 250 rpm; T: 25 °C; TMP: 1.4 bar	10.0	16.2	135	2776	The highest performance for the clarification of pomegranate juice was achieved for 0.05% of Al_2_O_3_-integrated PSF/ PEI membranes with the highest total soluble solid (16.2 ± 0.0 Brix), color (5781 ± 4 PtCo), and total phenolic content (2642.1 ± 46.4 mg GAE/L).	[2]
Sugarcane	CM; TMP: 9.3 bar; Temp: 25 °C; Time: 1 h	6.66	18	0.13 10.85	17.13 16.86	The cake layer build-up on the surface of the UF membrane was observed to be the main fouling trends, and could be addressed via physical and chemical cleaning, thereby improving the general efficiency of the filtration process.	[32]
Grapefruit	PES-MF; A: 17 cm^2^; T: 25 °C; TMP:0.5 & 1.5 bar	9.9	NA	2	20	In optimal conditions, the permeate flux of immersed membranes configuration attained 5 Lh^−1^.m^−2^.	[33]
Watermelon	PES-UF; MWCO:50 kDa; TMP: 0.5–3 bar; A: 50 cm^2^; pH:2–13; Temp: 30 °C	7.1	6.9	NA	NA	Significant reductions in color (3.16–0.245% A420); turbidity (2.11–83.17% T660); lycopene: 33.51–12.15 mg/L. The ascorbic acid content in the permeate was on the lower side than in the feed.	[34]

**Table 2 membranes-13-00679-t002:** Summary of findings of membrane fouling mitigation strategies during fruit juice filtration.

Nature of Juice	Membrane Type and Operating Conditions	Nature of Fouling Mitigation Technique	Major Findings	References
Brown sugar redissolved syrup	CM; Crossflow; TMP: 4 bar; cross-flow velocity: 4 m/s	Polydopamine	Modified ceramic membrane exhibited enhanced permeation flux of 193.75 LMH and higher turbidity reduction (˃99%).	[35]
Apple	PVDF-MF; Crossflow; A: 22.8 cm^2^; TMP: 0.3 bar; cross flow rate: 229.3 mL/min; T: 25 °C	Polydopamine coating and nisin	Pure water flux increased from 480.8 to 491.4 Lm^−2^h^−1^, nisin-grafted membrane showed greater performance in terms of hydrophilicity and anti-bacterial property, and 14.6% less decline in flux was also attained.	[36]
Pomegranate	PSF/PEI-MF; Dead-end; A:14.6 cm^2^; stirring speed: 250 rpm; T: 25 °C; TMP: 1.4 bar	TiO_2_ and Al_2_O_3_	The 0.01% TiO_2_ membrane had the highest pure water flux of (2776 L/m^2^h) with the least contact angle (69°) as compared with 94° for a non-modified membrane.	[2]
Apple	PSF/PEI-UF; cross-flow; A: 150 cm^2^; TMP: 5.4 bar; Time: 120 & 150 min.	Al_2_O_3_ and TiO_2_	Membrane modified with 0.01% TiO_2_ recorded the highest apple juice flux of 44.6 L/m^2^h, superior porosity, and hydrophilicity.	[30]
Pomegranate	RO-PSF; TMP: 3000 kPa; Temp: 25 °C;	Low-pressure nitrogen plasma	Lower contact angle (13.2°), increased flux, and soluble solids content for 90 W at 15 min.	[37]
Apple	UF-PSF; A: 0.0140 m^2^; TMP: 250 kPa; flow rate: 210 L h^−1^; Temp: 25 °C	Low-pressure oxygen plasma treatment	Higher hydraulic permeability (36.1–152 Kgm^−2^s^−1^ kPa^−1^) × 10^5^ under 90 W plasma power and 10 min exposure time.	[38]
Sugar cane	PSF-UF; TMP: 104 kPa; Flow rate: 30 L/h; Time: 8 h; Temp: 25 °C	Polypyrrole and chitosan composite	Increased flux permeability from 9 to 16.3 L/m^2^.h; 71.2% rejection of polyphenol oxidase enzyme, 17% flux recovery, and 76% reduction in membrane hydraulic resistance.	[26]
Sugar cane	CM; Temp (60,75,90 °C); pH (7.2, 7.5, 7.8); A: 0.1193 m^2^; TMP: 0.26 MPa	Lime saccharate	The pre-treatment increases the cake resistance. Partial liming of the juice at 75 °C produces higher permeate flux even at VCF of 20. Almost 90.17% purity and increased flux (121–248 L/m^2^.h) were achieved.	[39]
Cranberry	UF; Time: 60, 120 min; MWCO (50, 100, 500 kDa); A: 0.014 m^2^; Temp: 15 °C.	Pectinolytic enzyme	Depectinization for 60 and 120 min reduced UF duration by 16.7 and 20 min, improved the permeate fluxes, and reduced the duration of clarification.	[40]
Opuntia cactus cladode	MF-PES; A: 3.14 cm^2^; TMP: 0.2 bar; Time: 20 min	Pectinex Ultra, SP-L, Viscozyme-L	Viscozyme-L decreased the soluble polysaccharide content and attained lower viscosity and better membrane performance. The cake layer was the dominant resistance to membrane fouling during filtration.	[41]
Carrot	PVDF; A: 78 cm^2^; P:1000 W, frequency: 20 kHz; distance: 1.7 cm; flow rates (10, 15, 20 mL/s); TMP: 0.5 & 1 bar; Temp: 25 °C	Ultrasound treatment	The best performance in terms of turbidity rejection (97.9%, particle size 531.1 µm, increased permeate flux, and excellent feed flow rate were recorded at 1 bar and 15 mL/s).	[42]
Banana	UF; A: 0.0032 m^2^; TMP: 2 bar; enzyme concentrations: (0.1, 0.2, 0.3, 0.4, and 0.5%)	Pectinase enzyme (pre-treatment)	Pre-treatment of the banana juice before ultrafiltration also improved the permeate flux by 65.5% compared to the untreated sample.	[43]

## 2. Factors Affecting the Performance of a Membrane during Fruit Juice Clarification

The level and frequency of membrane fouling during juice clarification and concentration are affected by several factors, such as the nature of the membrane, juice composition, as well as filtration conditions (including transmembrane pressure (TMP), temperature, and cross-flow velocity (CRV)). Figure 1 illustrates the factors that could influence the performance of the membrane during fruit juice clarification.

### 2.1. Nature of Membrane

Membrane type and its properties (for example, configuration, hydrophobicity, and pore size) considerably influence the membrane performance during juice clarification and concentration [44], with respect to permeate flux, fouling propensity, cost, preservation of desired substances, as well as facile cleaning procedure [38,45,46,47]. Selecting a suitable membrane for a particular product can considerably lower fouling, hence enhancing process performance. A membrane with smaller MWCO or lower pores is usually closely linked with lower permeate flux as well as less fouling during separation [38,48]. However, a membrane with higher MWCO (more porous) can hold more particles, triggering acute pore blocking. For instance, Le et al. [19] investigated the influence of pore sizes (5, 10, and 20 kDa) under different TMP (1, 2, and 3 bar) on the clarification of red-fleshed dragon fruit juice using PSF-UF membrane. The result of their findings showed that membranes with 10 kDa pore size recorded the highest permeate flux (nearly 7.9 Kgm^−2^h^−1^) at 3 bar. The authors stated that resistance analysis showed that cake resistance (69–94%) was the main factor reducing flow and the main source of fouling in the UF process. Similarly, Zhu et al. [49] conducted a study to assess the effect of membrane pore size (20, 30, 50, and 100 kDa) on membrane fouling and filtration kinetics of UF membranes. Their findings indicated that the 20 kDa membrane exhibited the slowest filtration kinetics and higher insulin loss (almost 70%), while the 50 kDa membrane had better protein removal with inulin loss (nearly 15%). Ilame and Singh [50] examined kinnow juice clarification and established that the 30 kDa membrane exhibited the maximum permeate flux, owing to higher pore-clogging in the 44 kDa membrane, whereas the pores of the 10-kDa membrane were too tiny to navigate feed having colloidal particles.

Importantly, the membrane material is another critical factor for juice clarification and concentration, subject to the purposes. Some of the frequently used materials include inorganic materials (such as metals and ceramics) and organic polymers such as poly-ether sulphone (PES), polyamide (PA), cellulose acetate (CA), cellulose esters, polypropylene (PP), polysulphone (PS), polyvinylidene fluoride (PVDF), and polyacrylonitrile (PAN) [51]. Generally, membranes with hydrophobic characteristics are more vulnerable to fouling. For example, Nourbakhsh et al. [52] stated that almost 46% of red plum juice was developed, and more cake resistance was recorded using MF hydrophobic PVDF membrane in comparison with a hydrophilic mixed cellulose ester (MCE) membrane. The hydrophobic property of PVDF repels water, hence hampering feed from surging into membrane pores and accelerating cake development. Correspondingly, Gulec et al. [45] assess the clarification of apple juice and membrane fouling using various UF membranes. Their study revealed that hydrophobic membranes with rougher surfaces exhibited more severe fouling.

Several membrane configurations comprising hollow fiber, tubular, plate-and-frame, flat sheet, and spiral wound are frequently applied in the fruit juice processing industry. The tubular membrane exhibited comparatively minimal fouling propensity owing to the turbulent flow produced in addition to its larger inner diameter, which permits clarification (filtration) of juice with bigger particles, elevated solids content as well as greater viscosity [33]. De Oliveira et al. [53] observed that the tubular ceramic MF membrane had higher permeate flux and better juice quality compared with the hollow fiber PA membrane during the clarification of fruit juice. The turbulent flow produced in the tubular membrane improved particle diffusion, causing a lesser amount of cake accretion.

### 2.2. Temperature

Temperature control during juice clarification is highly essential to achieve desired product quality as well as optimal process efficiency. This is because a rise in working temperature typically lowers the viscosity of juice as well as enhances its diffusion coefficient along with permeate flux, causing minimal membrane fouling [54]. Contrastingly, Campos et al. [55] reported that the permeability flux of grape juice in MF as well as UF declined when the temperature was increased from 30 to 40 ℃ under 3 bar (TMP) as a result of the clogging of membrane pores at elevated pressure. Despite the enhancement in permeate flux, most juice clarification systems are performed under a temperature range of 20–30 ℃, not often above 55 ℃, since elevated temperature may cause amplified nutritional and sensory degradation, augmented microbial development, elevated energy demand, as well as destroy membrane materials.

### 2.3. Juice Composition

The constituent of juice is a significant element that must be characterized before settling the operational condition of a filtration procedure. Generally, fruit juice comprises various constituents, including polysaccharides (such as starch, cellulose, pectin, lignin, and hemicellulose), proteins, lipids, and polyphenols, which influence the juice density, viscosity, as well as concentration, and play an essential part in membrane fouling [19]. According to Darcy’s law, the permeate flux of a filtration system is inversely proportionate to the viscosity of the feed stream [38]. Numerous reports stated that feed streams with lesser viscosity caused minimal membrane fouling [56]. Zhao et al. [57] observed that raising the pectin concentration in apple cider developed more colloidal cloud particles with proteins as well as polyphenols, which enhanced membrane fouling as well as feed viscosity.

### 2.4. Transmembrane Pressure

Membrane filtration occurs when permeate is impelled by transmembrane pressure (TMP) to navigate via the membrane. Initially, permeate flux is controlled by TMP during filtration, then later impacted by concentration polarization when the controlling flux is achieved, and finally, the fouling layer begins to cake. Nonetheless, TMP has no effect on the permeate flux under a steady state [58]. For instance, Mai [54] reported that the permeate fluxes of UF for dragon fruit, cashew apple, pineapple, and pomelo juices were enhanced with rising TMP until attaining the regulating values of 1.5, 3, 1.5, as well as 2 bar, respectively. Though, a further rise in TMP lowered or stabilized permeate fluxes due to the rapid accretion of foulants on the surface of the membrane, resulting in severe fouling. Nguyen et al. [59] reported that the permeate flux was found to rise from 57.72 L h^−1^m^−2^ to 92.10 L.h^−1^/m^2^ due to the increased applied TMP (2–5 bars). Omar et al. [27] have confirmed the effect of operating pressure on the permeate flux character, the fouling mechanisms, and the juice quality properties such as turbidity, pH, color, ascorbic acid content (AAC), total phenolic content (TPC) as well as total soluble solids (TSS) during the clarification of guava juice using UF membrane system in a batch study. A higher operating pressure resulted in elevated flux during the UF process, with a limiting flux (J_lim_) found at 17.22 kg/m^2^/hr. The author revealed that both total and intermediate pore-clogging had been detected as the main fouling mechanisms in the process. Almost 97% turbidity reduction with a 17% TSS reduction (7–17%) was attained compared to the fresh juice. Though, 18 to 22% and 19 to 27% reductions of AAC and TPC, respectively, were observed in the clarified juice in relation to the fresh juice.

### 2.5. Cross-Flow Velocity

The shear produced through tangential feed flow over the membrane surface substantially impacts permeate flux and membrane fouling [51]. Mirsaeedghazi et al. [60] observed a decline in foul resistance and enhanced permeate flux during MF clarification of bitter orange as cross-flow velocity enhanced due to a greater tangential force applied by the feed. Bevilacqua [14] also reported that the elevated shear stress produced by raising cross-flow velocity could mitigate concentration polarization and eliminate foulants on the membrane, thus enhancing the permeate flux of watermelon juice during the UF process. However, Polidori [3] reported a decline in permeate flux as concentration progresses for both fresh and processed orange juices. This phenomenon is very typical in cross-flow MF. Based on the generalized Darcy law, it is the outcome of the increase of retentate viscosity as well as its fouling ability, which directly contributes to the overall hydraulic resistance of the system.

## 3. Strategies for Curtailing Membrane Fouling to Enhance the Fruit Juice Processing Performance

Generally, fouling reduces the permeate flux and enhances the flow resistance during filtration; efficient mitigation approaches for increasing the filtration performance have become the principal focal point of juice processing. Given these, this section discusses recent trends in various strategies employed for membrane fouling mitigation during juice clarification studies. These include membrane modification, juice pre-treatment, membrane cleaning, ultrasound, and hydrodynamics. Figure 2 highlights the strategies for enhancing membrane performance and fouling mitigation during fruit juice processing.

### 3.1. Membrane Modification

In late 2005, scientists uncovered nanoparticles with hydrophilic properties and investigated the interactions between polymeric membranes and nanoparticles. Nanoparticles have outstanding hydrogen bonding as well as electrostatic interaction, which stimulates the binding with a polymeric material [61,62]. In this context, transforming the membrane surface is among the most efficient strategies to enhance the anti-fouling properties by strengthening the hydrophilicity. To subdue the fouling difficulty, the membrane surface perhaps transformed using hydrophilic nanofillers and polymers, for instance, polyvinyl pyrrolidone (PVP), poly ethylenimine (PEI), polyethylene glycol (PEG), polyvinyl alcohol (PVA), low-pressure nitrogen plasma, low-pressure oxygen plasma, polypyrrole and chitosan, TiO_2_, Al_2_O_3_, polydopamine, and others. Among these hydrophilic polymers, PEI is a favored modifying agent due to its pore-forming potential [63]. However, the pore developer potential of PEI can trigger a reduction in mechanical potency as well as membrane selectivity [64].

Plasma-treatment techniques present highly developed platforms for a speedy functionalization of materials, permitting the concurrent tuning of surface morphology as well as the energy of the materials. Plasma treatment routes are flexible and eco-friendly technology that has also been practical across various macro-porous membrane materials utilized for UF or MF to enhance the efficiency as well as the anti-fouling character of the membranes, using different reactant gases. Furthermore, plasma surface alteration is among the most appealing technology, given that the process is dry and comparatively undemanding to regulate. Since it employs only gas as a reactant to modify membrane surfaces evenly and rapidly, gas plasma treatments can be considered a superior and neater technological option for membrane surface modification [65]. Gulec et al. [38], in their experimental study, modified the surface of the commercial PSF/UF membranes with low-pressure nitrogen plasma to study the influence of the exposure time and plasma power on the wettability and morphology of PSF membrane and evaluate plasma-treated membrane performance for clarification of raw apple juice in batch concentration system. After their experiment, it was observed that the hydraulic permeability of the modified membrane increased from (36.1 to 152 Kgm^−2^s^−1^kPa^−1^) × 10^5^) under 90 W (plasma power) within 10 min exposure time.

Bagci et al. [37] investigated the effect of low-pressure nitrogen plasma activation on the surface properties of a TFC composite polyamide RO membrane. A typical declining trend was noticed for the contact angle with rising exposure duration, attaining a minimum of 13.2 ± 0.8° for 90 W within a period of 15 min. During the RO filtration, an outstanding upsurge in water flux of the modified RO membrane was noticed, which raised elevated soluble solids content (SSC) values in the concentrated pomegranate juice. The most significant benefit of plasma treatment, specifically in membrane applications, is that the surface properties of the membrane could be transformed devoid of undermining its bulk structure [66]. Though, the result of the plasma activation treatment may vary based on the type of juicing substrate as well as the reactant gas in connection with the plasma variables, including treatment duration, excitation power, as well as pressure [67]. However, the major shortcomings of plasma treatment technology are that the molecular tuneability and chemical homogeneousness cannot be precisely established.

Several studies have been conducted to address the fouling drawback by integrating nanomaterials into the membrane dope matrix. Alteration of the membrane with nanomaterials enhances selectivity, tensile strength, permeability, as well as chemical and thermal resistance of PSF/PEI membranes [68,69]. Various studies have reported that nanofillers can enhance membrane permeability, hydrophilicity, antifouling, and porosity by modifying different polymeric membranes [70]. Membrane modification with nanomaterials facilitates enhanced permeability, selectivity, thermal and tensile strength, as well as chemical resistance [71]. Table 1 shows the summary of findings regarding fruit juice concentration strength and permeates flux during membrane clarification processes. For instance, Severcan et al. [34] investigated and developed a nanocomposite PSF ultrafiltration membrane modified with TiO_2_ and Al_2_O_3_ using a phase inversion procedure to enhance the clarification of apple juice and improve the apple juice flux. The PSF/TiO_2_/Al_2_O_3_/PEI membrane demonstrates outstanding pure water flux and anti-fouling resistance. Particularly, the 0.01% TiO_2_ membrane recorded the highest pure water flux of 2776 L/m^2^h under 5.4 bar with the least contact angle (69°) compared to 94° for non-modified membranes. The surface pores on the PSF/TiO_2_/Al_2_O_3_/PEI membrane are increased compared to those on the neat PSF membrane. Similarly, Severcan [2] also recorded a superior apple juice flux of 44.6 L/m^2^h with enhanced porosity and hydrophilicity with 0.01% TiO_2_ modified membrane when TiO_2_ and Al_2_O_3_ were used to modify PES/PEI UF membrane during apple juice clarification. Hence, the membrane hydrophilicity was enhanced, preventing the agglomeration of foulant on the membrane surface.

Hubadillah et al. [31] experimented to clarify kelime, watermelon, and kiwi fruit juices and enhance juice permeation as well as fouling behaviour during the metakaolin-based ceramic filtration process. Their findings reported a pure water flux of 273 L/m^2^.h and excellent mechanical strength (176.8 MPa). Xiong et al. [35] recently developed anti-sugar-juice fouling and temperature-resistant membrane surfaces by coating the surface of the ceramic membrane with polydopamine (PDA) during the clarification of brown sugar redissolved syrup. The result of their study shows that the modified ceramic membrane exhibited an enhanced permeation flux of 193.75 LMH and higher turbidity reduction (˃99%). They also stated that incorporating a PDA coating layer can efficiently improve the hydrophilicity of the membrane. Panigrahi et al. [26] prepared an antifouling and antimicrobial PSF membrane by blending polypyrrole and chitosan composite to address the fouling during the clarification of sugarcane juice using UF. A 0.5 MHCPY incorporated membrane exhibited an excellent increased flux permeability from 9 to 16.3 L/m^2^.h; 71.2% rejection of polyphenol oxidase enzyme, 76% reduction in membrane hydraulic resistance, as well as 17% flux recovery. It is worth noting that membrane surface transformation can also be accomplished via covalent organic frameworks (COFs) as well as metal–organic frameworks (MOFs), being two new types of porous crystalline materials, with excellent potential in membrane separations. These materials exhibit numerous similar structural properties, enabling them to reveal similar characteristics and corresponding applications [72]. Recently, MOFs and COFs novel materials are of broad interests owing to their remarkable chemistry and prospective applications, especially as fillers in both mixed-matrix membranes and thin film nanocomposite membranes. However, their application for modifying membrane surfaces to enhance selectivity and antifouling performance during fruit juice clarification is limited. Figure 3 illustrates various structural transformation techniques in enhancing membrane properties and performances.

### 3.2. Pre-Treatment of Juice

#### 3.2.1. Chemical or Enzymatic Pre-Treatment

Since polysaccharides are considered the principal constituent in juice triggering membrane fouling, they require the pre-treatment of juice with certain enzymes, including pectinase, to lessen its viscosity, thereby increasing its filterability. Adding pectinase before juice clarification efficiently enhances the permeate flux together with decreased fouling growth [40].

A combination of pectinases, along with other enzymes, including proteases, amylases, as well as cellulases depending on the nature of the juice constituent, can be employed to disintegrate more juice components that support fouling [78]. Given this, Shi and his co-workers [39] examined the effect of pre-treatment using lime saccharate on juice quality permeate flux and fouling during the clarification of sugar cane juice using a ceramic membrane (pore size of 20 nm). Their findings revealed that the permeate flux increased from 160 to 278 L/m^2^.h at a pH of 7.2, with the highest impurity reduction. Their study concluded that the pre-treatment of the juicing substrate increases the cake resistance, while partial liming of the juice at 75 °C was responsible for the higher permeate flux recorded even at VCF of 20. Similarly, Perreault et al. [40] recently assessed the influence of the depectinization process using pectinolytic enzymes as a pre-treatment before clarifying cranberry juice on the performance of the UF process and the cranberry juice constituent. Higher permeate fluxes and reduction in clarification period were achieved due to depectinization. Depectinization for 60 and 120 min reduces UF filtration duration by 16.7 and 20 min. The superior filtration performance, with respect to permeate fluxes, was achieved with the 500 kDa UF membrane despite the highest total flux decrease (41.5 to 57.6%). The authors concluded that the fouling layer at the membrane surface was comprised of anthocyanins and polyphenols.

Yee et al. [43] investigated the influence of enzymatic pre-treatment on the permeate flux behavior, fouling mechanism, and juice quality using UF for banana juice clarification. Their findings indicated that pectinase-treated juice exhibited a higher permeate flux rate than the untreated juice during the UF clarification, with a reduced filtration time by half. The authors revealed that cake formation is the major cause of membrane fouling.

#### 3.2.2. Physical Pre-Treatment

This technique comprises sedimentation and coagulation, which can also mitigate membrane fouling during juice clarification [79,80]. In comparison to the chemical approach/enzymatic hydrolysis, the physical pre-treatment technique is ideal, particularly for juice dehydration via osmosis, because these techniques permit uncomplicated juice alteration by re-introducing the filtrated juice constituents to the permeate [80].

Ultrasound is an unconventional physical approach for fouling mitigation during membrane separation. The ultrasonic modification has been observed to lessen cake layer accretion, increase permeates flux, and enhance juice cloud stability during MF clarification of pomegranate juice [81] as well as carrot juice [42,82]. Aghdam et al. [81] stated that ultrasound treatment lowered the overall fouling resistance in pomegranate juice clarification using MF, with cake resistance and irreversible fouling resistance. Similarly, Hemmati et al. [42] investigated the influence of ultrasound at (1000 W, 20 kHz) on the permeation flux and during MF membrane filtration of Carrot Juice. The author observed that the permeate flux was increased by almost 10^−3^ Kg/m^2^s at a TMP of 1 bar and a flow rate of 15 m/s. The rate at which the ultrasound-induced flux increases is directly proportional to the rise in transmembrane pressure (TMP) and the flow rate of the feed. The study’s findings confirmed that ultrasound combined with a membrane unit enhanced the permeate flux during the clarification of carrot juice. However, the phenolic compounds and total soluble solid content remain unchanged throughout the clarification process.

Expanding ultrasound frequency at steady output power perhaps produces more cavitation bubbles, though with tinier size; thus, the cavitational collapse is less intense, causing reduced shear force and turbulence to decrease fouling. Consequently, the augmentation of membrane separation performance and the efficient cleaning of fouled membranes using ultrasound applications have been established by several studies [81,83,84]. Generally, it was unearthed that ultrasound was effective for enhancing permeate flux and reducing membrane fouling [85]. Gao et al. [86] observed an increase in the normalized permeate flux from 0.21 without ultrasound to 0.7 with ultrasound under a power of 16 W and frequency of 20 kHz for a cross-flow UF system. Though expanding ultrasound power can efficiently abate membrane fouling, not all membranes can be enhanced with elevated ultrasound power owing to or subject to their mechanical stability. For instance, Hou et al. [87] examined hollow fiber membranes used in an ultrasonic-aided membrane distillation system. Their study indicated that the permeate flux increased from 5% to 60% based on the operating conditions at lesser feed velocity and temperature, small frequency, higher ultrasonic power, and elevated feed concentration. Their study also stated that the ratio of the permeate flux with ultrasound to that with zero ultrasound can be expanded. Cho et al. [88] employed ultrasonic cleaning on fouled membranes, and they observed that it is more effective in eliminating membrane foulants and regaining the flux comparison with conventional physical and chemical cleaning techniques. However, the authors reported structural damage to the membranes when the use of elevated ultrasonic power at low frequencies (150 and 300 W at 28 and 45 kHz). Flat sheet membrane configuration can withstand elevated ultrasound power in comparison with hollow fiber membranes. Additionally, superior ultrasound power cannot be applied to PS hollow fiber membranes with no support layer [89]. Another drawback of the ultrasonic technique is that the resultant increasing temperature may adversely affect the physicochemical characteristics of fruit juice, which needs to be controlled by further temperature control. Physical centrifugation of juice constituents comprising large suspended particles (for instance, fruit pulp) can efficiently decrease juice turbidity and color, reducing membrane fouling during filtration [59].

#### 3.2.3. Hydrodynamics

Being a significant component in cross-flow membrane separation, alteration of surface tangential shear from a hydrodynamic viewpoint perhaps regulates the transport trends of particles as well as solute away from the surface of the membrane. Thus, each membrane configuration must be designed by identifying the interconnection between local mass transfer and hydrodynamic shearing to curtail concentration polarization of solute in addition to inhibiting foulant accretion [90]. For instance, Xie et al. [91] reported the effect of micro-channel turbulence promoters on the hydrodynamic efficiency of submerged membrane bioreactor. The experimental result and computational fluid dynamic simulation indicated that the average gas velocity, wall shear stress, turbulent kinetic energy, and average fluid velocity equipped with MCTP in the vertical orientation are above the horizontal orientation with 11.26%, 15.79%, 37.76%, and 8.7%, respectively.

Tsai et al. [92] developed the 3D turbulence promoters printing technique (with diamond, elliptic, and circular configurations) for cross-flow MF to study the effect of TMP, cross-flow velocity, and geometry of the promoters on cake properties and permeation flux. Their findings indicated that the filtration flux increased by almost 155% due to the increase in cross-flow velocity from 0.1 to 0.5 m/s/. It was also observed that the elliptic promoter with a hydraulic angle (90) recorded the flux enhancement by (30–64%) under 20 kPa (TMP) compared with normal MF. Their study concluded that incorporating turbulence promoters could efficiently diminish cake build-up on the membrane surface. Recently, research by Dattabanik et al. [93] incorporated simple wire-type turbulence promoters of dynamic shear to enhance fouling mitigation during the stimulated UF filtration process for protein recovery from food wastewater, resulting in a 445% enhancement of permeate flux.

#### 3.2.4. Membrane Cleaning

Membrane cleaning for fouled membranes can be conducted through various methods, broadly categorized into two categories: physical and chemical cleaning. The physical cleaning technique alters the applied different temperatures or turbulence together with the hydrodynamics of the membrane system to dynamically drive the foulant to remove from the membrane. Contrastingly, chemical cleaning involves using chemicals to modify the solution chemistry and altering the electrical dual layer to facilitate electrostatic interaction between the foulants and the membrane [62]. A chemical cleaning procedure for UF membrane is a generally accepted technique aimed at lessening flux loss caused by irreversible fouling. Chemical cleaning of the membrane with irreversible fouling can be accomplished by using chemical agents like alkalis, detergents, acids, as well as oxidants. However, hypochlorite (NaOCl) persists in a prominent selection due to its availability, inexpensive, produces a reduced amount of harm to the membrane, and ability to impede fouling through a satisfactory cleaning process [94].

Chemical cleaning can assist in regaining the initial performance of the membrane and evacuating the foulants. In the meantime, uninterrupted chemical cleaning leads to membrane structural degradation, resulting in a reduction in its mechanical strength, hydraulic performance, and physical as well as chemical structures. It is important to identify the cause of the decline and enhance the lifespan of the membrane. Researchers use NaOCl cleaning agents due to their cleaning capacity not favoring organic foulants, flexibility, and stability with other cleaning agents.

Generally, both physical and chemical cleaning processes are utilized jointly to increase the performance of the cleaning process. However, there are weaknesses and challenges in utilizing chemical and physical cleaning. These include achieving an adequate circulation of water flow during the membrane process, careful study of the particle removal mechanism, impacts of the period for water sparging and the velocity of the water on the removal performance, and the nature of the membrane system [23,95,96].

In recap, membrane modification, which involves transforming the membrane surface properties to be more hydrophilic, remains an efficient technique for mitigating fouling challenges. However, further studies are required to explore the performance of polymeric membranes using various other supplements comprising covalent organic frameworks (COFs), metal–organic frameworks (MOFs), and nanomaterials. Also, membrane modification in combination with other physical pre-treatment techniques could offer excellent fruit juice clarification with better fouling-resistant properties.

## 4. Conclusions and Future Research Recommendations

The processing of fruit juices has remained a significant component in the food processing industries. The inherent drawbacks associated with the conventional thermal evaporation techniques as well as increasing demand for high-quality fruit juice with minimal or no impact on nutritional properties have motivated various studies to consider environmentally benign membrane technology owing to its unique, promising properties. UF and MF membrane processes are the most widely exploited processes using PSF and PES polymeric materials. In the fruit juice industry, the potential benefits of UF and MF processes over other filtration processes are irrefutable owing to enhanced product quality and reduced energy demand. Fruit juices obtained by membrane filtration have outstanding quality. However, membrane fouling remains the major problem constraining the application of membrane technology since it shortens the lifetime of the membranes. Membrane modification, cleaning, juice feed pre-treatment, ultrasound, and hydrodynamics are recent strategies that could mitigate fouling in fruit juice clarification. For future research, a combination of any of the aforementioned techniques could be further explored. Finally, there is a need to incorporate more hydrophilic nano-supplement such as MgO, Fe_3_O_4_, ZnO, SiO_2_, Zeolites, Ag, carbon nanotubes (CNTs), etc., in the membrane matrix to strengthen further the membrane properties and selectivity in tackling fouling. Although a growing number of studies focusing on membrane modification through nanomaterials are being reported, there is a lack of knowledge regarding the application of MOFs and COFs materials for membrane modification for fruit juice clarification, together with the interactions between MOFs/COFs and membrane processes during fruit juice clarification. However, studies have indicated that MOFs/COFs are highly promising materials capable of considerably improving membrane performance [72]. Hence, further studies need to be conducted to improve the knowledge of how MOFs/COFs can mitigate membrane fouling during fruit juice filtration to overcome the fouling issue.

## Figures and Tables

**Figure 1 membranes-13-00679-f001:**
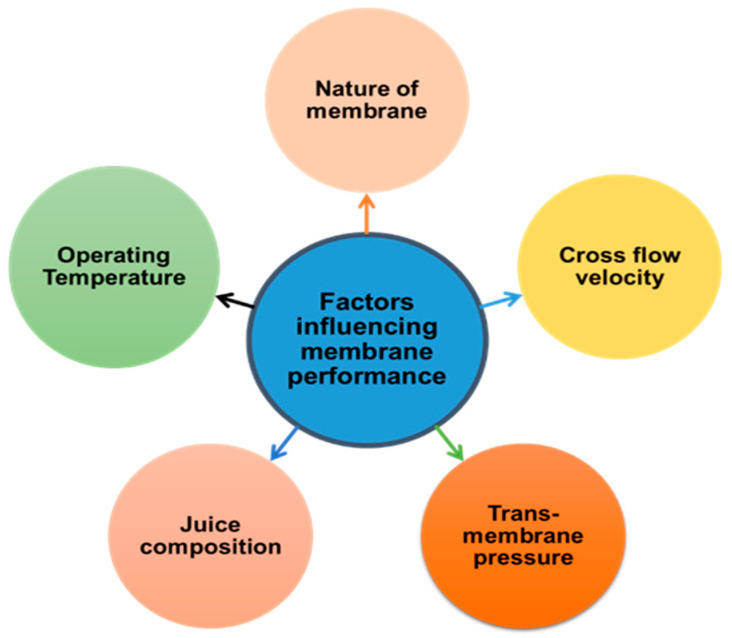
Factors controlling membrane performance during fruit juice clarification.

**Figure 2 membranes-13-00679-f002:**
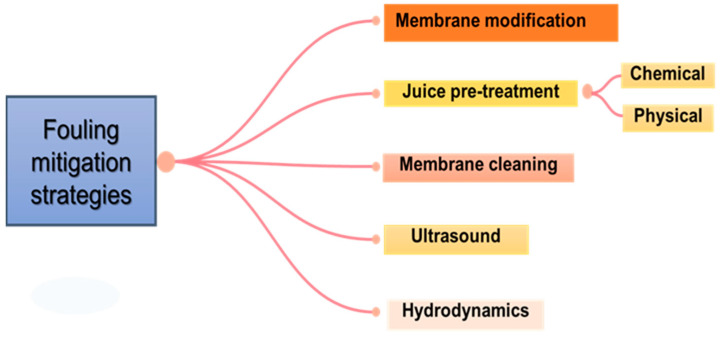
Strategies for enhancing membrane performance and mitigating fouling during fruit juice clarification.

**Figure 3 membranes-13-00679-f003:**
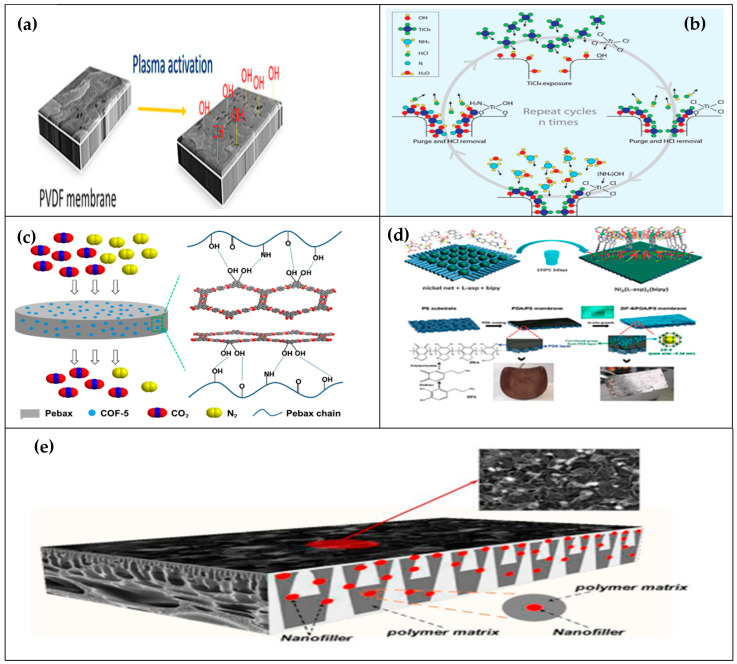
Membrane structural transformations via plasma activation [73] (**a**); nitrogen-doped-activated membranes [74] (**b**); covalent organic frameworks modified membrane [75] (**c**); metal–organic frameworks modified membrane [76] (**d**); and nanofiller modified membrane [77] (**e**).

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
