# Peer review of "Strategies to Enhance the Membrane-Based Processing Performance for Fruit Juice Production: A Review"

_membranes, 2023, doi:10.3390/membranes13070679_

Round 1
Reviewer 1 Report
The proposed article deals with an interesting theme with important scientific and application issues. However, in my opinion, the proposed synthesis contains a lot of generalities already relatively well known and described. It does not seem to me really original compared to the current state of the art. In the end, the analysis is based on a rather small number of examples, whereas the literature offers a plethora of them. Finally, the document contains errors in the units and some quick approximations on certain notions. Therefore, as it stands, the article does not seem to me to bring enough added value to be published in the journal Membranes.
Author Response
Dear sir,
Attached herewith is the point-by-point responses.

Reviewer 2 Report
A comprehensive and well structured review of the topic, with relevance for the fruit juice processing industry.
Author Response
Dear sir,
The responses are hereby attached herein.

Reviewer 3 Report
In general, it is a good review that reviews recent advances regarding strategies to improve the performance of membrane technology in fruit juice processing. Although the review is thorough and pertinent, some specific points need to be improved. Some specific comments in this regard are the following:
lines 75,- 83: a series of technological alternatives are listed for the improvement of the quality of juices through non-thermal processes, but with a single reference. It is suggested to further enrich the references in this paragraph.
lines 99 – 100: is it easy to scale-up a membrane process? This phrase is often repeated in more than one article, but in practice the scaling-up of a process is not linear because aspects such as the geometry of a laboratory module, compared to an industrial process. It can be very different, bringing important hydrodynamic effects. Please check this section of the review more critically.
Section 3.1 is the part of the manuscript that provides the greatest novelty regarding the state of the art. It would be appropriate to have a diagram or graphical representation of what happens in membranes that are subjected to different forms of structural modification such as addition of plasma treatments, low-pressure nitrogen activation, nanofillers, MOFs, and COFs.
Lines 479-490: only one paper is cited that refers to the use of MOFs, in a circumstance that discusses the effect of ultrasound on membranes during a filtration operation. This is also confusing because this discussion is part of a section that mentions physical solution pretreatment as a fouling mitigation strategy. One thing is the pretreatment itself, another is the ultrasound-assisted membrane filtration. It is requested to clarify this paragraph and incorporate relevant references.
lines 149-157: review the wording of this paragraph and its elements: such as flux units, is it inulin or insulin separation? What do you mean by the increased resistance of filtration cake from 69-94%? Respect of what? What are those filtration kinetics parameters that are alluded to?
lines 471-475: what do you mean that the flux increased due to the increase in feed flow rate and the TMP due to the ultrasound? One thing is the increase in flux due to the operating condition, and another is the result that the ultrasonic cavitation effect has caused prior to membrane filtration in the solution. Please Clarify this part because it is very confusing as it is.
Author Response
Dear sir, attached herewith is the responses to the reviewers comments.

Reviewer 4 Report
Dear Authors,
Basically, I like this work on this topic very much. Also the presentations and the structure with regard to the overview of the data. Also the subject matter, which is rarely published in this form in individual works. In my opinion, a large part of the expertise lies with companies that deal with this kind of product reprocessing, but do not publish anything beyond that, with the exception of patents.
A list of abbreviations would be very helpful, making it easier for the reader to find abbreviations that are not commonly known.
In reviews, however, all experiences that have been published to date must be described and the work in which these experiences were gathered must be cited in the most precise manner. This is also in order to be able to read about this and that new and interesting experience, how it was obtained.
Some things seem to have got mixed up with the literature citations and the numbering. A check of all citations and allocations is absolutely necessary.
I followed up on some of them and repeatedly found literature that did not correspond and then gave up.
Further hints and remarks will follow:
Lines 78-82: What the reference 4 has to do with this description of concentration processes without heat treatment is not quite clear to me. It would be very desirable to provide relevant and meaningful literature on all the different processes mentioned.
Line 88-90: In the widespread applications of membrane technology in combination with fruit juices, desalination is certainly a given in terms of its economic efficiency, but I do not find it suitable for wastewater recycling. There are many applications in pharmaceuticals and chemical production where this is completely true, but not in wastewater, where membranes in end-of-pipe solutions do not seem sustainable to me.
Line 90: That this absolute property of membranes is demonstrated here with a review [6] is questionable for me here as well.
Line 116: The presentation of the superior selectivity, the low-cost film formation potential, the excellent mechanical properties, the high chemical and thermal resistance and the low energy requirement can be shown with the literature citation [17]. Is this literature applicable here?
In general, these descriptions of superlatives are very close to an assessment and should be proven here by applicable treatments, if at all that is possible!
Lines 124-126: The hydrophobic properties of PSF and PES polymer membranes are referred to here and the literature reference given deals with ceramic membranes [18] !?.
130: The term "anomaly" does not seem to be correct for these effects.
Line 142: The literature [22] is cited for the references to permeate flow, fouling tendency, costs, retention of the desired substances and ease of cleaning. Are these references also adequately addressed here with the work dealing with "a Comparative Evaluation of Membrane Fouling and Juice Quality"? After all, this is only about apple juice clarification.
Line 144: Is the literature dealing with "Performance enhancement of ultrafiltration in apple juice clarification by low-pressure oxygen plasma: a comparative evaluation versus pre-flocculation treatment". relevant for “ … membrane with smaller MWCO or lower pores, lower permeate flux as well as less fouling during separation”. From the title of this cited work [23], I cannot understand this.
Line 166: Ref.: [27] stated that almost 46% of the red plum juice was developed and more cake resistance was recorded..... In contrast the paper [27] (line 701) deals with the physico-chemical properties of ultrafiltrated kinnow fruit juice (mandarin). Again, I cannot find any correspondence between the text and the literature citation.
Line 206: Numerous reports have found that feed streams with lower viscosity cause minimal membrane fouling [33]. To this end, in the cited paper [33] (line 714) reports on the sustainable dewatering of grapefruit juice by forward osmosis, wherein the improvement of membrane performance, fouling control and product quality was the objective. Again, it is not clear to me why the influence of viscosity on fouling can generally be derived from investigations with the special FO for pressure-driven membrane processes.
Line 391: ...the highest pure water flow of (2776 L/m2h)... Why is this value in brackets and the pressure applied would also be required for all the flow data?
There are other flow values given in the text, where the pressure applied is also not given. Specific flow values are usually in L/m2/h/bar.
There are other flow values given in the text, where the pressure applied is also not given. Specific flux values are usually in L/m2/h/bar, or you can also write this with brackets, e.g. L/(m2hbar), or also as Lm-2h-1bar-1.
In line 387, however, the cited literature is given as "Severcan et al., [40]". This literature by Severcan et al. is not [40] in the index but [56] in line 767.
Author Response
Dear sir,
Find the attachment as regards responses to reviewers comments.
Thank you.

Round 2
Reviewer 3 Report
The quality of the work has been improved by far from the former version, and it results attractive and a contribution that properfly fits the standards of this journal.
Author Response
Thank you for your comments. All observations were duly noted.
Reviewer 4 Report
Dear Authors,
It would be a pity if this work could not be published, but in doing so, all citations should be checked that their attribution is correct.
I picked out two literature citations and both were incorrect:
Line 197: …Ilame & Singh [38] examined kinnow juice clarification… However, in the bibliography, [38] it is “Mai, H. C. 2017..."
Line 479; …Panigrahi et al., [57][69] prepared an antifouling and antimicrobial PSF membrane…, but in the bibliography, under item 69. a paper by Zhang et al. is cited.
I checked one more in line 212 likewise not correct.
In the case of citations in the tables and texts, a name should also be given next to the number from the bibliography. This would also be a control for checking the literature assigned in each case.
Until the literature is fully cited correctly here, I cannot recommend this work for publication.
A list of abbreviations would be very helpful, making it easier for the reader to find abbreviations that are not commonly known and is still missing, or I could not find it.
Author Response
Thank you for all your comment.
Responses has been provided in the attachment. A list of abbreviations has been provided in the supplementary file.

Round 3
Reviewer 4 Report
Dear Authors,
I would like to thank you for making the necessary changes, especially in the citation of literature, and I think that this paper will also meet with great interest on this subject.